

# Resolving nanoparticle growth mechanisms from size- and time-dependent growth rate analysis

Lukas Pichelstorfer[1], Dominik Stolzenburg[2], John Ortega[3], Thomas Karl[4], Harri Kokkola[5], Anton Laakso[5], Kari E.J. Lehtinen[5,6], James N. Smith[7], Peter H. McMurry[8], and Paul M. Winkler[2]

[1]Division of Physics and Biophysics, Department of Materials Research and Physics, University of Salzburg, Salzburg, Austria
[2]Faculty of Physics, University of Vienna, Vienna, Austria
[3]Atmospheric Chemistry Observations and Modeling Laboratory, National Center for Atmospheric Research, Boulder, Colorado, USA
[4]Institute for Meteorology and Geophysics, University of Innsbruck, Innsbruck, Austria, University of Innsbruck
[5]Finnish Meteorological Institute, Atmospheric Research Centre of Eastern Finland, Kuopio, Finland
[6]Department of Applied Physics, University of Eastern Finland, Kuopio, Finland
[7]Department of Chemistry, University of California, Irvine, California, USA
[8]Department of Mechanical Engineering, University of Minnesota, Twin Cities, Minneapolis, Minnesota, USA

*Correspondence to:* Paul M. Winkler (Paul.Winkler@univie.ac.at)

**Abstract.** Atmospheric new particle formation occurs frequently in the global atmosphere and may play a crucial role in climate by affecting cloud properties. The relevance of newly formed nanoparticles depends largely on the dynamics governing their initial formation and growth to sizes where they become important for cloud microphysics. One key to the proper understanding of nanoparticle effects on climate is therefore hidden in the growth mechanisms. In this study we have developed and

successfully tested two independent methods based on the aerosol general dynamics equation, allowing detailed retrieval of time- and size-dependent nanoparticle growth rates. Both methods were used to analyze particle formation from two different biogenic precursor vapors in controlled chamber experiments. Our results suggest that growth rates below 10 nm show much more variation than is currently thought and pin down the decisive size range of growth at around 5 nm where in-depth studies of physical and chemical particle properties are needed.

**1  Introduction**

Aerosol nanoparticle formation from gas-to-particle conversion occurs frequently throughout the global atmosphere (Kulmala et al., 2004). Despite their small sizes these particles might be of climate relevance through the indirect aerosol-cloud effect (Twomey et al., 1984). Modelling results suggest that this secondary aerosol formation mechanism contributes roughly 50 % of particles to the budget of cloud condensation nuclei (Spracklen et al., 2008; Merikanto et al., 2009; Gordon et al., 2017).

New particle formation (NPF) has been subject to numerous studies for several decades. Besides experimental studies under ambient and laboratory conditions, substantial effort has been put into the modelling of aerosol dynamics to address phenomena such as nucleation, condensation/evaporation and coagulation. In order for newly formed particles to eventually become CCN, particles need to grow sufficiently fast to prevent them from being scavenged by pre-existing particles.



Importantly, the formation rate at a specific diameter $J(d_p)$ is highly sensitive to the diameter growth rate $dd_\mathrm{p}/dt$. Knowledge of $dd_\mathrm{p}/dt$ not only is needed to calculate particle formation rates but intrinsically contains information on the growth mechanisms (McMurry and Wilson, 1982). The diameter growth rate as a function of particle size and time is therefore key to the understanding of growth mechanisms during gas-to-particle conversion. Several authors have characterized growth rates

from the first appearance of various particle sizes over time, which is referred to as the appearance time method (e.g. Kulmala et al., 2013; Lehtipalo et al., 2014; Tröstl et al., 2016). However, this method cannot fully resolve size- and time-dependencies of the observed growth rates in highly dynamic systems. Therefore, several attempts have been made in the past to derive $dd_\mathrm{p}/dt$ by solving the general aerosol dynamics equation (GDE) (Lehtinen et al., 2004; Verheggen and Mozurkewich, 2006; Kuang et al., 2012) using growth rate analysis on the basis of experimental number-size-distribution measurements (Heisler

and Friedlander, 1977; McMurry et al., 1981; Wang et al., 2006). Those techniques might suffer from insufficient data quality, which is usually limited over a certain size range and/or time resolution of the sizing technique being applied. Number-size-distribution measurements typically take 1-2 minutes per scan, and can therefore be too slow to characterize the observed size-resolved growth rates. Hence, determination of $dd_\mathrm{p}/dt$ is still a major source of uncertainty in the proper characterization of nanoparticle growth. Here we present a new approach to this problem that compares two different methods based on GDE

analysis. The methods are tested and compared to simulated NPF events. Both approaches are then applied to experimental data from particles formed from the ozonolysis of monoterpenes and sesquiterpenes in a 10 m$^3$ aerosol chamber. State-of-the-art particle sizing instrumentation (Stolzenburg et al., 2017) enables the methods to quantify size- and time-dependent growth rates over the crucial size-range between 2-50 nm. This thereby helps to improve our understanding of the differences between monoterpenes and sesquiterpenes in new particle formation, which were emphasized by recent studies (Zhao et al., 2017).

## 2   Description of growth rate determination

Change rates of the number-size-distribution are described by the continuous GDE as in Seinfeld and Pandis (2006):

$$\frac{\partial n(v,t)}{\partial t} = \frac{1}{2}\int_0^v K(v-q,q)n(v-q,t)n(q,t)\,dq - n(v,t)\int_0^\infty K(q,v)n(q,t)\,dq - \frac{\partial}{\partial v}[I(v)n(v,t)] + S(v) - R(v) \qquad (1)$$

where $n(v,t)$ represents the number-volume-distribution at time $t$ and volume $v$, $K(v,q)$ is the coagulation kernel, $I(v)$ a particle current across the volume $v$ and $S(v)$ and $R(v)$ are size-dependent source- or remove-terms, respectively.

In a well controlled aerosol chamber experiment, the GDE is governed by just a few effects. An aerosol dynamics module accounting for dilution, wall losses and coagulation is used to calculate simulated number-size-distributions $n^\mathrm{sim}(t_j, d_p)$ (Pichelstorfer and Hofmann, 2015) evolving from measured experimental input number-size-distributions $n^\mathrm{exp}(t_{j-1}, d_p)$ between two subsequent time steps $t_{j-1}$ and $t_j$. With growth as the only unknown in the GDE, comparison between the simulated and the measured number-size-distribution $n^\mathrm{exp}(t_j, d_p)$ allows for its quantification. Here we focus primarily on two different

methods that have been employed to determine size- and time-dependent growth rates from this comparison. A brief description of the software tool used to interpret the experimental data is given below, details can be found in Appendix A and Appendix B. Potential errors of the analysis methods are discussed in Appendix F.



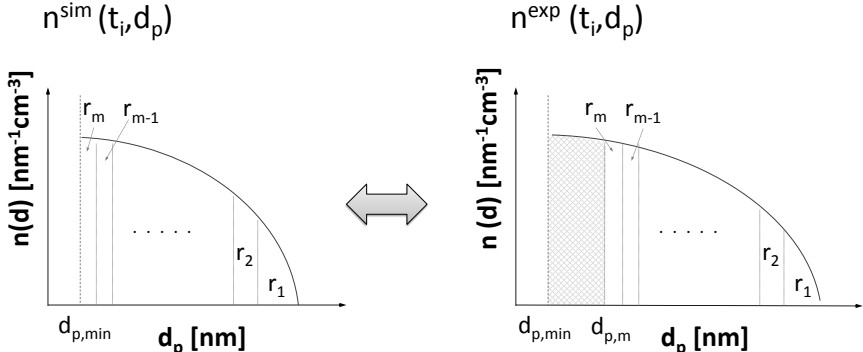

**Figure 1.** Schematic comparison of defined regions $r_{i=[1,m]}$ of the simulated $n^{\mathrm{sim}}(t_j, d_p)$ (left) and experimental $n^{\mathrm{exp}}(t_j, d_p)$ (right) number-size-distributions allows for the determination of the growth rate. Particle diameter is plotted on the abscissae, particle number-size-distribution is plotted on the ordinates of the graphs. The shaded area on the right-hand-graph depicts the particles that grew beyond the minimum diameter $d_{\mathrm{p,min}}$ within the last time span $\Delta t = t_j - t_{j-1}$. Particles of size $d_{\mathrm{p,min}}$ at time $t_{j-1}$ are of size $d_{p,m}$ at time $t_j$ (see right hand graph).

### 2.1 *T*racking *RE*gions of the *N*umber-size-*D*istribution: the TREND method

The first method in estimating particle growth rates is based on the assumption that regions $r_i$ between $d_{p,i}$ and $d_{p,i+1}$ (each containing a certain fraction $1/m$ of the total simulated particle number concentration $N^{\mathrm{sim}}_\infty$) of the simulated number-size-distribution, $n^{\mathrm{sim}}(t_j, d_p)$, can be assigned to regions in the experimental number-size-distribution, $n^{\mathrm{exp}}(t_j, d_p)$, (see Fig. 1).

Hence, the method tracks regions of the number-size-distribution, and is thereafter called TREND method.

The particle number concentration within each region $N_r$ is defined as

$$N_r = \frac{N^{\mathrm{sim}}_\infty}{m} \qquad \text{with} \qquad N^{\mathrm{sim}}_\infty = \int\limits_{d_{p,min}}^{\infty} n^{\mathrm{sim}}(t_j, d_p)\, dd_p \tag{2}$$

where $m$ is an integer parameter which determines the number of used regions and $d_{\mathrm{p,min}}$ is the smallest diameter used (e.g. a lower measurement threshold, or the size of a nucleating cluster). Note that the particle number concentration $N_r$ is

always determined from the simulated number-size-distribution $n^{\mathrm{sim}}$. The limits of the regions are related to the number-size-distribution by

$$\int\limits_{d_{p,r_{i+1}}}^{d_{p,r_i}} n(t_j, d_p)\, dd_p = N_r \tag{3}$$

This equation can be solved for the simulated ($n^{\mathrm{sim}}(t_j, d_p)$) and the experimentally determined ($n^{\mathrm{exp}}(t_j, d_p)$) number-size-distributions by setting the upper integration limit to the maximum diameter of the distribution $d_{p,r_{\mathrm{imax}}}$ and subsequent

numeric integration towards smaller particle sizes until $N_r$ is reached. In this way, the next region limit $d_{p,r_{\mathrm{imax}-1}}$ is found and





the procedure is repeated until all limits of the $m$ regions are determined. Figure 1 illustrates the principle for determination of the $m$ regions for $n^{\mathrm{sim}}(t_j, d_p)$ and $n^{\mathrm{exp}}(t_j, d_p)$. For each of the regions $(r_{i=[1,m]})$ of the experimental and the simulated number-size-distribution, the count median diameter $d_{\mathrm{CMD},i}$ is determined and used to calculate the growth rate $GR$ of a particle with diameter $d_{p,i}^* = \frac{d_{\mathrm{CMD},i}^{\mathrm{exp}} + d_{\mathrm{CMD},i}^{\mathrm{sim}}}{2}$:

$$GR\left(d_{p,i}^*\right) = \frac{d_{\mathrm{CMD},i}^{\mathrm{exp}} - d_{\mathrm{CMD},i}^{\mathrm{sim}}}{t_{j+1} - t_j} \qquad (4)$$

Note that the procedure described above has limitations and benefits of which the most important are listed below:

1. Growth is decoupled from the other dynamic processes. Thus large relative changes in the region limits $d_{p,r_i}$ may cause errors.

2. Rapid changes in the growth rate require adequate time resolution of the experimental data as the result of the analysis method is a mean growth rate for the respective time interval.

3. Influence of the coagulation process by particles smaller than $d_{\mathrm{p,min}}$ can only be estimated.

4. The present method utilizes integral values to determine the growth rate. Thus local minima and maxima of the measured number-size-distribution (e.g. due to low particle concentration) may cancel out. However, this depends on the choice of the width of regions, which can be set for each analysis run.

## 2.2 *IN*terpreting the change rate of the *S*ize-*I*ntegrated general *D*ynamic *E*quation: the INSIDE method

The second method is based on explicit manipulation of the adapted, size-integrated GDE (see Eq. 1) which gives the change in integrated volume-concentration featuring a volume larger than $v_{\mathrm{eval}}$:

$$\left.\frac{dN_\infty}{dt}\right|_{v_{\mathrm{eval}}}^{\infty} = \left.\frac{dv}{dt}\right|_{v_{\mathrm{eval}}} \cdot n(v,t)|_{v_{\mathrm{eval}}} + \frac{dN_{\infty,coag}}{dt} - \int_{v_{\mathrm{eval}}}^{\infty} \beta_{\mathrm{wall}}(v)\, n(v,t)\, dv - \beta_{\mathrm{dil}} N_\infty \qquad (5)$$

where $t$ is time, $v$ is the particle volume, $n(v,t)$ is the number-volume-distribution and $v_{\mathrm{eval}}$ the smallest particle volume considered (not necessarily equal to the minimum measured volume $v_{\mathrm{p,min}}$). $N_\infty$ depicts the total integrated number concentration, from $v_{\mathrm{eval}}$ to $\infty$. Note that compared to Eq. 1, the loss-terms have been adopted for a chamber-experiment and the particle current $I(v)$ now represents the particle growth at the evaluation volume $v_{\mathrm{eval}}$.

The first term on the right hand side considers particles that grow into the range $[v_{\mathrm{eval}}, \infty]$; the second term considers number-volume-distribution changes due to coagulation; the third term describes losses at system walls; and the fourth term losses by dilution. Coagulation and wall losses are approximated by a comparison between the simulated number-distribution $n^{\mathrm{sim}}(t_{j+1})$ and the experimental number-distribution $n^{\mathrm{exp}}(t_j)$ at two discrete and subsequent points in time and for the considered interval $[v_{\mathrm{eval}}, \infty]$:

$$\left.\frac{dN_\infty^{\mathrm{sim}}}{dt}\right|_{v_{\mathrm{eval}}}^{\infty} = \left.\frac{dN_{\infty,coag}}{dt}\right|_{v_{\mathrm{eval}},\infty} - \int_{v_{\mathrm{eval}}}^{\infty} \beta_{wall}(v)\, n(v,t)\, dv - \beta_{\mathrm{dil}} N_\infty \approx \frac{N_{[v_{\mathrm{eval}},\infty]}^{\mathrm{sim}}(t_{j+1}) - N_{[v_{\mathrm{eval}},\infty]}^{\mathrm{exp}}(t_j)}{t_{j+1} - t_j} \qquad (6)$$



Therefore equation (5) can be rearranged:

$$\frac{dv}{dt}\bigg|_{v_{\text{eval}}} = \frac{\frac{dN_\infty}{dt}\big|_{v_{\text{eval}}}^\infty - \frac{dN_\infty^{\text{sim}}}{dt}\big|_{v_{\text{eval}}}^\infty}{n\,(v,t)|_{v_{\text{eval}}}} \tag{7}$$

The differential of the total change in number concentration is similarly approximated by the difference between the experimental number-distribution of two subsequent points in time, i.e.:

$$\frac{dN_\infty}{dt}\bigg|_{v_{\text{eval}}}^\infty = \frac{N_{[v_{\text{eval}},\infty]}^{\text{exp}}\,(t_{j+1}) - N_{[v_{\text{eval}},\infty]}^{\text{exp}}\,(t_j)}{t_{j+1} - t_j} \tag{8}$$

Due to these approximations of the differential expressions in Eq. 7, the number concentration at the evaluated volume $n\,(v_{\text{eval}},t)$ is expressed as $(n^{\text{exp}}\,(v_{\text{eval}},t_{j+1}) + n^{\text{exp}}\,(v_{\text{eval}},t_j))/2$.

Furthermore the volume dependence is replaced by diameter, i.e. $d_{\text{p}} = (\frac{6}{\pi} \cdot v_{\text{p}})^{1/3}$ and number-volume-distributions $n\,(v,t)$ are transformed to the measured quantity of $dN/d\log d_{\text{p}}\,(d_{\text{p}}) = \frac{2.303\pi}{2} d_{\text{p}}^3 \cdot n(v_{\text{p}},t)$. As a result the diameter growth rate at diameter $d_{\text{eval}}$ can then be given by

$$\text{GR}\,(d_{\text{eval}}, (t_{j+1} + t_j)/2) = \frac{N_{[d_{\text{eval}},\infty]}^{\text{exp}}\,(t_{j+1}) - N_{[d_{\text{eval}},\infty]}^{\text{sim}}\,(t_{j+1})}{t_{j+1} - t_j} \cdot \frac{1}{\frac{1}{2.303\pi d_{eval}^3}\frac{dn}{d\log d_p}\,(d_{\text{eval}}, (t_j + t_{j+1})/2)}. \tag{9}$$

For the INSIDE method the most important limitations and benefits can be summarized as follows:

1. The INSIDE method also features the aspects 1 to 3 of the TREND method.

2. It allows for determination of GR at pre-selected diameters while method 1 determines GR and $d_p$ based on the number $m$ of regions considered.

3. Fluctuations or scatter in the input number-size-distribution may significantly change the result due to the $\frac{dn(d_{\text{eval}},(t_j+t_{j+1})/2)}{d\log d_{\text{p}}}$ dependence.

## 3  Testing of the analysis methods

In order to test the analysis methods described above, number size distributions generated by the model SALSA [Sectional Aerosol module for Large Scale Applications; (Kokkola et al., 2008)] were used. Detailed information about the input parameters for the SALSA model can be found in the Appendix G. Figure 2 shows the growth rate functions serving as input for the SALSA model and the results of the two analysis methods.

Note that no fitting was done. Both models nicely capture the slope of the input growth rate curve, however, there are some deviations. Both models show an increasing scatter of the data with increasing particle diameter. This can be explained by the different representations of the number-size-distribution within the models. While the SALSA model uses a volume-based moving average representation, the analysis methods consider a distribution of the particles within each size bin. Thus the larger the particles grow the more pronounced the differences between the models become. Furthermore, some pronounced deviations





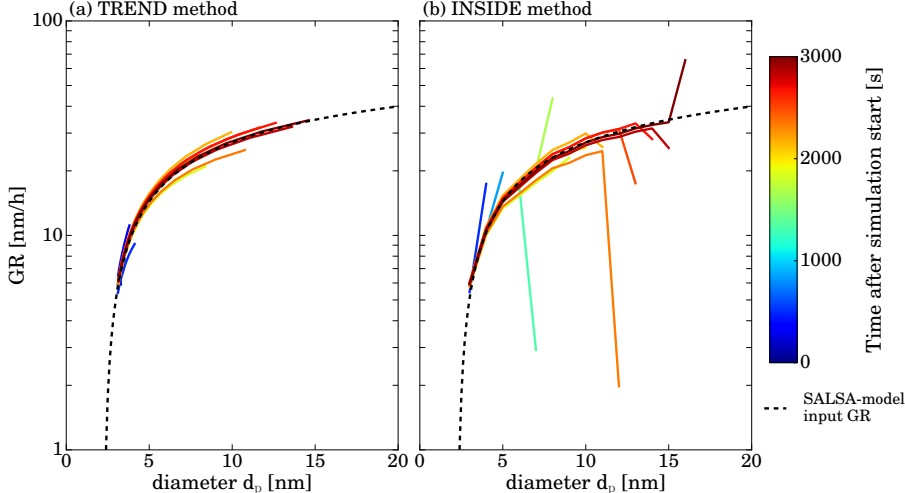

**Figure 2.** Results of the two GDE based analysis methods on simulated input size distribution generated by the SALSA model. Panel (a) shows the results of the TREND method and panel (b) the results of the INSIDE method. The black dashed line represents the time-independent input growth rate function, and the solid lines the results of the two methods as a function of diameter. The color code of the lines corresponds to the different times.

between measured and determined GR occur for the INSIDE method, which are not found for the TREND method. They show up only at the upper end of the number-size-distribution where number concentrations are low. These pronounced deviations are not found in the TREND method which uses integrated number concentration values with respect to dynamic diameters (see Fig. 1) to determine growth rates and hence all regions have fixed counting statistics. Except for this problem, both methods are able to determine growth curves by analysing $\frac{dN}{d\log d_p}$ representations of number-size-distributions especially for small particle sizes and low particle numbers. Statistical analysis of the deviation between the generated growth rate (SALSA) and results from the INSIDE- and TREND method reflect this behaviour. Each analyzed growth rate data-point is compared to the SALSA input value at the same diameter. The mean relative deviation and the corresponding standard deviation are 1.2 % and 5.0 % for the TREND method and 6.5 % and 12.7 % for the INSIDE method, respectively. The effect of higher particle concentration and hence larger influence of coagulation has been investigated in similar simulations featuring higher nucleation rates and are discussed in the Appendix C. It seems that the TREND method works better for analysing the leading edge of the newly formed particle size distributions and it in general shows less scatter due to its integral method. On the other hand, the INSIDE method performed very well when analysing GR at higher particle concentrations; however, it is more sensitive to scatter in the input data which has to be considered when real measured data is analysed.





## 4 Growth rate evaluation from chamber experiments

Both methods described above were used to analyse growth rates from NPF events produced in the aerosol chamber at the National Center for Atmospheric Research (NCAR) in Boulder, CO, USA. Experiments were performed in a 10m³ teflon bag which was continuously flushed by zero air at a flow rate of 40 lpm. A biogenic VOC ($\alpha$-pinene or $\beta$-caryophyllene) was

added to the zero air until steady state concentrations of $\sim$4 ppb were obtained. Subsequently, a UV mercury-lamp was turned on in one of the zero air lines to increase ozone in the chamber steadily and initiate ozonolysis of the VOC and subsequent NPF. For both experiments steady-state ozone concentrations of $\sim$25-30 ppb were reached at the end of the experiment. Additional details of the experimental set-up can be found in Winkler et al. (2013). Evolution of the number-size-distribution was monitored by a regular scanning mobility particle sprectrometer (SMPS) and a prototype differential mobility analyzer -

train (DMA-train). The DMA-train uses several DMAs and condensation particle counters (CPCs) in parallel. Each DMA is set to transmit only particles of a specific mobility diameter to monitor the size evolution of individual sizes preferably in the sub-10 nm size range at high time resolution (in the order of seconds). Operation principles from a similar, advanced setup can be found in Stolzenburg et al. (2017).

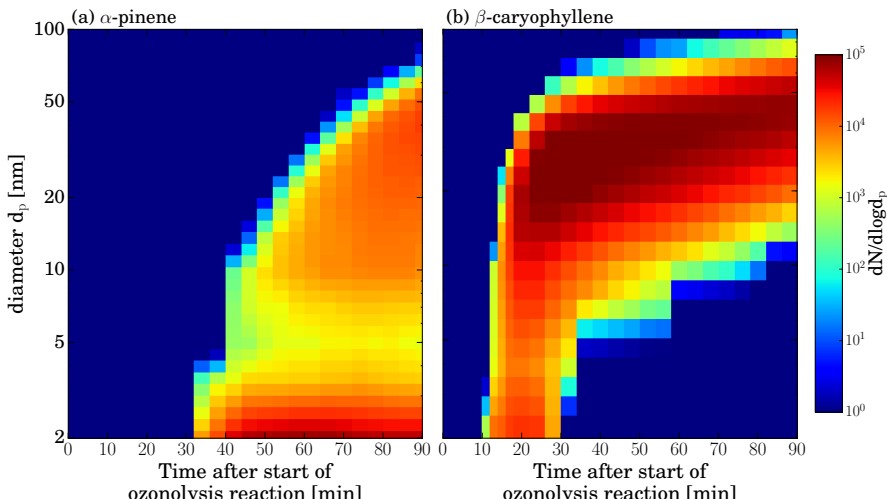

**Figure 3.** Combined, DMA-train and SMPS data, showing size distribution evolution over time for the ozonolysis of two different VOCs. Panel (a) shows the $\alpha$-pinene ozonolysis experiment and panel (b) the $\beta$-caryophyllene ozonolysis experiment. Transition between DMA-train and SMPS measuement is at 14 nm.

Figure 3 shows combined size distribution measurements for particles from both VOCs. Details of the data inversion proce-

dure can be found in Appendix E. It can be clearly seen that not only the absolute particle yield is higher in the $\beta$-caryophyllene system, but also growth proceeds much faster than in the case of $\alpha$-pinene. While for the sesquiterpene the first appearance of particles is observed after $\sim$10 minutes it takes roughly three times as long for the monoterpene. Obviously, there is quite different growth dynamics involved.




These different dynamics can be quantified by analysing the evolution of the number-size-distribution with the two above described methods. In Fig. 4 the results for the $\alpha$-pinene system are shown. Both methods show the same trend and similar absolute growth rate values. As already discussed during the test with simulated size distributions, the results of the TREND method do not cover the full size range at every time step due to the choice of the size interval number $m$. The INSIDE method

on the other hand generally shows more scatter, especially in regions where counting statistics above the evaluation size $d_{\mathrm{eval}}$ are poor and therefore those results are greyed out. This analysis reveals that growth rates above 10 nm have a negligible

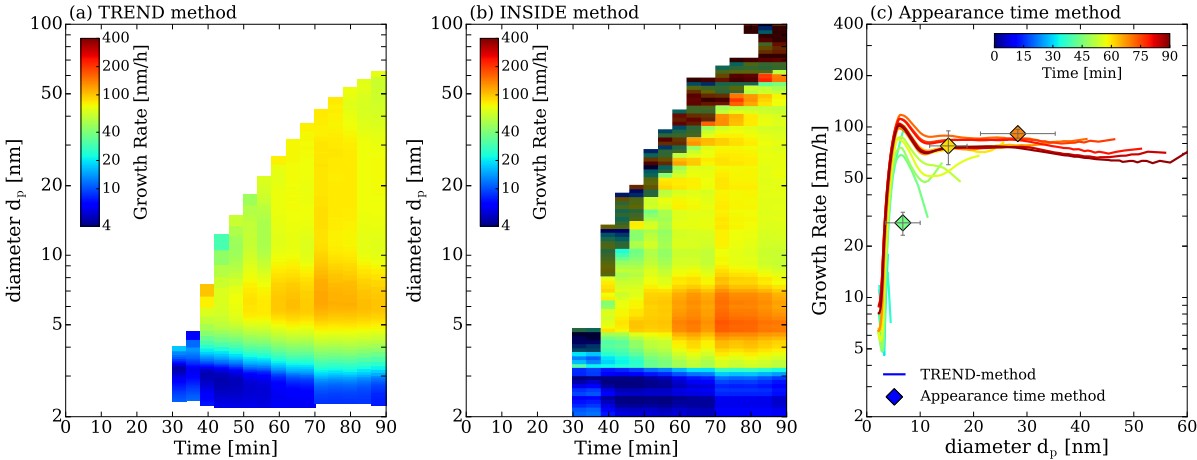

**Figure 4.** Growth rate analysis of the $\alpha$-pinene ozonolysis experiment. Panel (a) shows the results of the TREND method while panel (b) shows the results of the INSIDE method. The color coding represents the growth rates in nm/h. For the INSIDE method regions with low counting statistics are greyed out. Panel (c) shows a comparison of the growth rate analysis results obtained from the TREND method (continuous lines) with results from the widely used appearance time method (discrete points).

size-dependence. However, a strong size dependence is seen below 10 nm with peak growth rates around 7 nm and strongly decreasing growth rates in the sub-5 nm size range independent of the measurement time. This can be explained by a multi-component Kelvin effect, where some of the $\alpha$-pinene reaction products can only participate in growth when particles have

grown large enough to overcome the Kelvin-barrier, as shown in Tröstl et al. (2016) for the $\alpha$-pinene system. For the peak at 5 nm we can exclude the contribution of particle coagulation below the measurement size-range (Olenius and Riipinen, 2017) as shown in Appendix D.

Additionally, the results from the TREND method are compared with growth rate values calculated by the appearance time method in Fig. 4 (c). Good agreement is observed for all three possible appearance time method measurements. The

appearance time method does not reveal the complete time- and size-dependencies of the growth and could neither conclude about the observation of a multi-component Kelvin-effect nor about the observed higher growth rates at around 5 nm. Moreover the TREND method shows the clear trend of increasing growth rates until a more or less steady state growth is reached. We speculate that this is due to the slow accumulation of condensable low volatility vapours by the ozonolysis (proceeding at a





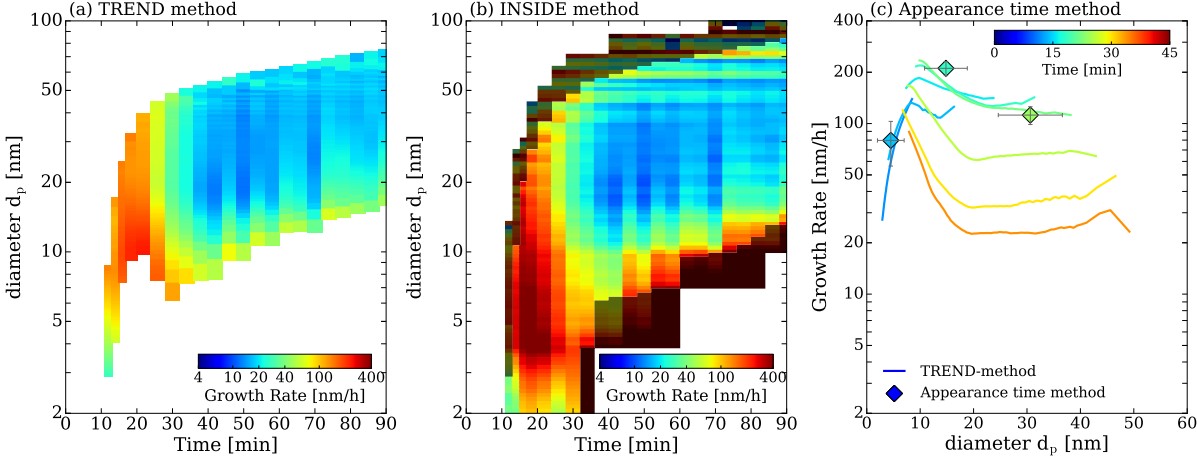

**Figure 5.** Growth rate analysis of the $\beta$-caryophyllene ozonolysis experiment. Panel (a) shows the results of the TREND method while panel (b) shows the results of the INSIDE method. The color coding represents the growth rates in nm/h. For the INSIDE method regions with low counting statistics are greyed out. Panel (c) shows a comparison of the growth rate analysis results obtained from the TREND method (continuous lines) with results from the widely used appearance time method (discrete points).

rate constant of $k_{\alpha\mathrm{p.*O_3}}(293K) = 9.06 \cdot 10^{-17}$ cm$^3$ molecule$^{-1}$ s$^{-1}$ (Atkinson et al., 2006)) and subsequent auto-oxidation of $\alpha$-pinene.

This seems to be completely different in the $\beta$-caryophyllene system. The results of the growth rate analysis are presented in Figure 5. The corresponding comparison of the results from the TREND method with the calculated growth rates from the appearance time method is displayed in Fig. 5 (c) and shows again excellent agreement. High growth rates at the beginning of the observed events are followed by a drop of growth rates in all sizes as the particle growth goes on. This can be explained by the very high oxidation potential and high reaction rates of $\beta$-caryophyllene, where the reaction rate constant for ozonolysis ($k_{\beta\mathrm{c.*O_3}}(298K) = 1.2 \cdot 10^{-14}$ cm$^3$ molecule$^{-1}$ s$^{-1}$ (Richters et al., 2015)) is three orders of magnitude higher than in the case of $\alpha$-pinene. Condensable vapours are therefore quickly formed and the steady state $\beta$-caryophyllene concentration at the beginning of the experiment is depleted by this fast reaction. Together with the fast build up of a large condensational sink, this shuts off new particle formation and reduces the fast growth rates.

In such a highly dynamic case it becomes evident that a higher time-resolution than the 240 seconds from the SMPS scans would yield a better dataset for the applied analysis methods. Additionally, when the particles reach larger sizes, the higher total particle number concentration increases the influence of coagulation and might disturb the results derived at small sizes. Moreover, due to the higher particle number concentrations in the growing mode, the inferred size-range of the growth rates by the TREND method shrinks. A more detailed discussion of the uncertainties of the two methods can be found in the Appendix F. In general, the biggest sources of errors are low time-resolution of the measurement data and scattering of the experimental data. Further, a source of high potential error is coagulation. For the experimental data presented in this work, the estimated





error of the GR determination associated with the analysis tools is typically in the range from 2 % to 35 % depending on the analysis method and the experiment.

Despite of the challenges in the highly dynamic case of $\beta$-caryophyllene ozonolysis, both methods reveal that the size-dependence of the growth rates is most significant in the sub-10 nm region, as in the case of $\alpha$-pinene. Moreover, the INSIDE method still covers the full size-range for the analysis of the size-dependence. When new particles are formed at the beginning of the experiment it reveals extremely high growth rates of up to 250 nm/h between 5 and 10 nm. Similar to the case of $\alpha$-pinene, but somewhat less significant, are the lower growth rates in the sub-3 nm range. This trend suggests that the condensable vapours produced from the ozonolysis of $\beta$-caryophyllene are mostly less volatile compared to the products of $\alpha$-pinene and can therefore participate in the growth from the smallest sizes on. This is predictable because a sesquiterpene with 15 carbon atoms will be less volatile than a monoterpene (e.g. Donahue et al., 2011).

## 5 Conclusion

We presented two methods to determine size- and time-dependent growth rates by analysing particle size distributions and solving the GDE. The TREND method tracks regions of the number-size-distribution. The INSIDE method is based on interpreting the size-integrated GDE, and determines growth rates at certain $d_{\mathrm{eval}}$.

Both methods reliably reproduce input growth rates from simulated size distributions and allow for quantitative comparison. The TREND method generally shows less scatter and less sensitivity to low counting statistics but cannot always cover the full range of particle sizes where growth is actually observed. The INSIDE method is capable of determining growth rates wherever particles are measured. However, determination of growth rates at very low or very high particle concentrations may suffer considerable errors. This is due to insufficient counting statistics of the measured input data on the one hand, and considerable coagulation effects on the other hand. While coagulation is typically considered in the GDE analysis a precise description of coagulation requires detailed knowledge of the aerosol properties (e.g.: inter particle forces or shape (e.g. Chan and Mozurkewich, 2001)) which are typically unknown for newly formed particles.

We applied our methods to experimental size distribution data from chamber studies to derive size- and time-dependent growth rates from ozonolysis of two different biogenic VOC precursors. Both methods agree well with the widely used appearance time method and provide valuable insights on some unexpected details of the growth dynamics in these systems.

For both studied VOC systems, a strong increase of growth rates was found for the smallest diameters until a maximum value is reached at around 7 nm. This finding strongly suggests that (biogenic) growth is governed by a multi-component Kelvin effect allowing condensation of vapor molecules only if the particles exceed a certain size. This observation is very pronounced in the case of a-pinene and was reported independently from other studies (Winkler et al., 2012; Tröstl et al., 2016). For the $\beta$-caryophyllene system, it is less significant, indicating that the major part of the products of $\beta$-caryophyllene ozonolysis are generally less volatile and can participate in growth at particle diameters well below 10 nm. This system showed a highly dynamic behaviour and fast changing growth rates over time, as the condensable vapours became quickly depleted in the chamber due to the high reactivity of $\beta$-caryophyllene and low volatility reaction products. Growth rates above 10 nm





generally showed only a minor size dependence. Regarding the different nanoparticle forming behaviour of monoterpenes and sesquiterpenes similar findings were recently reported from plant emission studies in a chamber environment (Zhao et al., 2017).

Our analyses underline the critical need to accurately quantify growth dynamics in the sub-10 nm size range. This range is
crucially important for the survival probability of newly formed particles and clearly features the biggest changes of growth rates. It is one of the prerequisites for the successful application of our newly developed methods to have size distribution measurements providing sufficient time-resolution and appropriate counting statistics. We see these requirements fulfilled in latest state-of-the-art instrumentation (Jiang et al., 2011a; Stolzenburg et al., 2017) allowing full exploitation of growth dynamics in the future. We also plan to make the analysis tool kit publicly available in order to allow for a wide application
and improvement by the scientific community.

*Author contributions.* P.M.W., J.O, T.K, P.H.M., J.N.S. performed the experiments, L.P, D.S, P.M.W., P.H.M. developed the analysis tools, D.S., L.P. analysed the data, H.K., A.L., K.E.J.L. provided the simulation input, L.P, D.S, P.H.M., K.E.J.L., P.M.W. were involved in the scientific interpretation and discussion, L.P., D.S. and P.M.W. wrote the manuscript.

*Acknowledgements.* This work was supported by the European Research Council under the European Community's Seventh Framework
Programme (FP7/2007–2013)/ERC grant agreement No. 616075. P.H.M.'s work was supported by U.S. DOE grant DE-SC0011780. The National Center for Atmospheric Research is supported by NSF.




## Appendix A: Description of the software tool used to interpret the experimental data

The flowchart contained in Figure 6 outlines the data analysis method. In the first step, number-size-distributions measured or generated by means of computer simulation is transformed from $\frac{dN}{d\log d_\mathrm{p}}$ representation to bin concentration. This includes an automatic fitting process since the incoming data provides information at given diameters (i.e. no analytical function). The representation of the particle size distribution is

5  similar to the so-called hybrid structure (Chen and Lamb, 1994) and features a fixed size grid containing uniform distributions each having an upper and a lower limit and a number density (i.e. a particle number concentration per diameter interval). Integration of number density from the lower to the upper size limit results in the number concentration within the bin. This structure allows for calculation of coagulation and phase transition without suffering from numerical diffusion. Furthermore, it provides a continuous-like number-size-distributions which is required to minimize numerical error in the growth rate calculation. A more detailed description can be found elsewhere (Pichelstorfer and Hofmann, 2015).

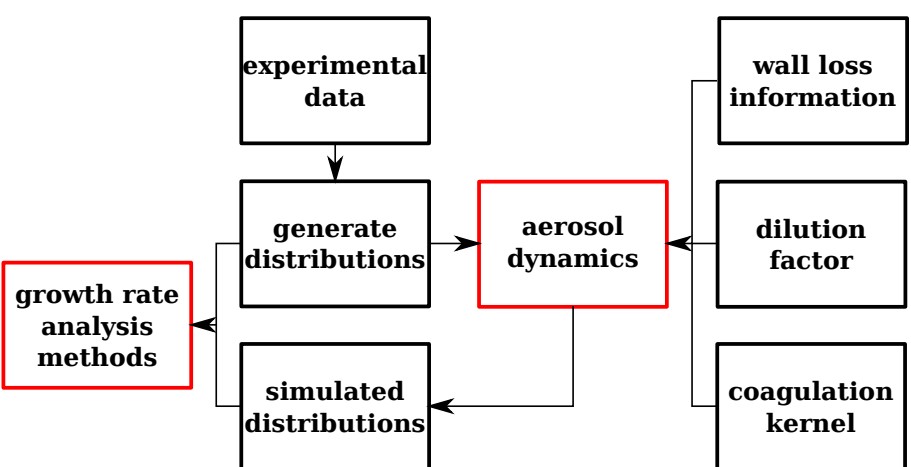

**Figure 6.** Flow chart describing the principle of the data analysis

The input (experimental or simulated) number-size-distribution (at time $t_j$) and wall loss information are used to calculate dynamical changes to the aerosol that occur during the period of time between two measurements. These changes include coagulation as well as deposition and dilution. Note that the influence of particles smaller than a minimum diameter $d_\mathrm{min}$ is not considered in this simulation. Details of the aerosol dynamics module can be found in Appendix B. The result of the aerosol dynamics simulation is a simulated particle

15  size distribution at time $t_{j+1}$ which is then compared to the measured particle size distribution at time $t_{j+1}$ to determine the growth rate.



## Appendix B: Description of the aerosol dynamics module

Figure 7 outlines the procedure of the aerosol dynamics module. An experimentally determined particle size distribution $n_{exp}(t_j, d_\mathrm{p})$ measured at time $t_j$ enters the integration time loop (ordinary Eulerian forward integration). The aerosol altering processes, namely dilution, coagulation and deposition are calculated sequentially. A control parameter $C$ is used to ensure that relative changes done to the distribution

are below a certain maximum value (e.g. 0.1 % relative change in particle concentration within a time step) to enable quasi-simultaneous calculation of the processes. If the change is larger than this limit, changes during this integration time step are ignored and the integration time step $\Delta t$ is divided by two. Otherwise the distribution is updated and system time $t_i$ is increased by $\Delta t$. That way integration time steps are optimized in order to save computational time and achieve desired accuracy. The result of the aerosol dynamics module is a simulated dis-

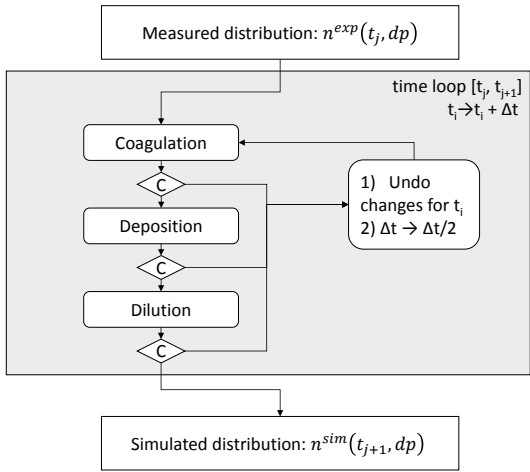

**Figure 7.** Flow diagram of the aerosol dynamics model calculating the changes to a particle size distribution $n_{exp}(t_j, d_\mathrm{p})$ within the time interval $[t_j, t_{j+1}]$. $C$ stands for control parameter and $t$ stands for time.

tribution $n_{sim}(t_{j+1}, d_p)$ at time $t_{j+1}$. This distribution was generated neglecting the influence of nucleation, coagulation of particles smaller

than $d_{\min}$ as well as phase transition, respectively. Coagulation is described by numerically solving a discrete version of the Smoluchowsky equation (Smoluchowski, 1917):

$$\frac{dn_k}{dt} = -\sum_{i=1}^{s} \beta_{ik}(..)n_i n_k + \frac{1}{2} \sum_{f(i+j)=f(k)} \beta_{ij}(..)n_i n_j \tag{B1}$$

where $n_k$ is the number concentration within size bin $k$, $t$ is time and $\beta_{ik}$ is the coagulation coefficient describing the probability of two particles of size bin $i$ and $k$, respectively, to collide with each other. Collisions are assumed to be 100 % effective. Furthermore, the

15 only coagulation mechanism considered is thermal coagulation of neutral (i.e. uncharged) particles. A potential error caused by neglecting additional inter particle forces is discussed and estimated in Appendix F. A more detailed description of the solution can be found elsewhere (Pichelstorfer et al., 2013).





Wall loss of particles is described by:

$$\frac{dn_i}{dt} = -\beta_{wall,i} \cdot n_i \tag{B2}$$

where $\beta_{wall,i}$ is the size-dependent wall loss coefficient determining the loss of particles of size $i$ per second. In the present work the wall loss rate is obtained from literature for particles larger than 12 nm (Fry et al., 2014). For smaller particles, the loss rate was estimated based

5    on experimental data using a method described by Crump and Seinfeld (1981).

Dilution is described similar to wall loss. For the description of dilution we assume that the chamber is well mixed (i.e. no concentration gradients, which has been verified in the NCAR chamber using $CO_2$ tracer experiments). Thus, dilution can be described as analogues to wall loss by applying a size-independent loss coefficient, which can be determined from dilution flow and chamber volume.

The result of the simulation is a particle size distribution $n_{sim}(d_{\mathrm{p}}, t_{j+1})$, which is calculated based on the experimentally determined

10    number-size-distribution at time $t_i$, $n_{exp}(d_{\mathrm{p}}, t_j)$.





## Appendix C: Performance in case of high particle concentrations

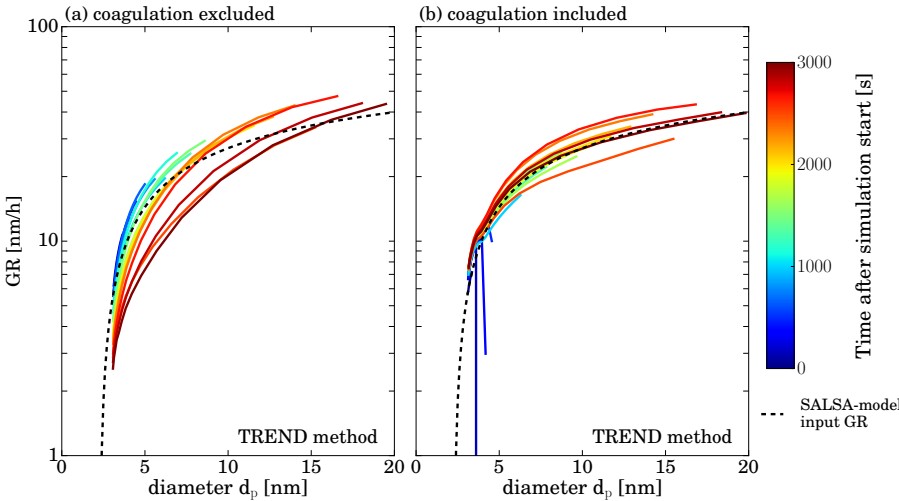

**Figure 8.** Analysed (straight) and set growth rate (dashed) as a function of particle diameter. Panel B shows results obtained by considering the effect of coagulation while results depicted by Panel A where determined neglecting coagulation.

In order to test the models' performance in case coagulation has to be considered similar simulations featuring higher nucleation rates ($J = 3000$ particles $\cdot$ cm$^{-3}\cdot$ s$^{-1}$) were considered.

Figure 8 depicts analysed GR and set growth rate, respectively, against particle diameter for two different simulations using TREND
5 method. Panel A neglects coagulation while panel B considers coagulation. Clearly Panel B shows better agreement between the theoretical curve and the analysis result for small particle diameters. As particles get larger, the data points start to scatter. This is caused by the different representations of the particle size distributions by the analysis software and SALSA model. However, note that equations used to calculate the coagulation kernel are the same for both models.





## Appendix D:  Potential effect of particles below the detection limit on the growth rate

As described in Section 2.2 both methods of the developed growth rate analysis do not take into account the interaction with particles smaller than the size detected experimentally (i.e. below 3 nm in diameter). This effect is known to cause difficulties in aerol dynamics simulations (Olenius and Riipinen, 2017). While it is rather unlikely that those small particles affect the deposition of larger particles to the chamber walls

significantly, they might cause additional particle growth due to coagulation. Neglecting this contribution might cause an overestimation of the growth rate.

In order to estimate the effect of particle coagulation of sub-detectable sizes on the growth rate, we first generate a particle size distribution based on McMurry (1980). McMurry considers the formation rate of condensable monomers by chemical reaction and their subsequent growth to larger particles by coagulation. In this, dimensionless equations were solved numerically to generate dimensionless number-size-

distributions. The dimensionless number-size-distributions used in this work where determined similarly to this method but using a different model representation of the number-size-distribution (McMurry and Li, 2017) and converted to a dimensional representation by using the following parameters (adjusted for oxidation products of $\alpha$-pinene, according to Kirkby et al. (2016)):

  – Particle density of 0.5 g cm$^{-3}$ to 2 g cm$^{-3}$

  – Monomer volume of 0.2 to 0.8 nm$^3$ (based on the density and an estimated molecular mass of 246 g/mol for the condensable vapor

– Monomer formation rate of $4.4 \cdot 10^4$ s$^{-1}$ cm$^{-3}$ (estimated based on O$_3$ and $\alpha$-pinene concentration considering a reaction constant of $k_{\alpha\mathrm{p}.*\mathrm{O}_3}(278K) \approx 4.0 \cdot 10^{-17}$ cm$^3$ molecule$^{-1}$ s$^{-1}$ (Atkinson et al., 2006) and a highly oxidised molecules (HOM) formation probability of 2.9 % at 278 K for the reaction product (Kirkby et al., 2016))

The resulting number-size-distributions are depicted by Fig. 9. Obviously, concentrations in the detectable size range (i.e. larger than 3 nm for the given experiments) suggested by the numerical method are much higher than measured ones. This might be caused by the fact

that our basic approach to this numerical method neglects wall losses. However, we concluded that the number-size-distributions determined may still be used to estimate a maximum contribution of sub-3 nm particles to the growth rate.

The number-size-distribution in the diameter range from 0.8 nm to 3 nm was divided in five logarithmically spaced sections. In the next step the contribution to the growth rate per hour of a particle due to coagulation with particles (constant concentration of 1 p/cm$^3$ ) in each of the sections is computed. The coagulation kernels are determined using a formulation for the transition regime (e.g. Hinds, 1999).

Multiplication of this growth rate function by the number-size-distribution as determined above results in the growth rate [nm/h] of particles larger than 3 nm due to coagulation (see Fig. 9 (b)). For the sake of completeness we also plotted the effect considering monomer addition (i.e. condensational growth). The maximum effect on growth is around 1.7 nm/h for a particle density of 2 g/cm$^3$ and around 1.2 nm/h for a particle density of 0.5 g/cm$^3$. Comparing this numbers to Fig. 4 and 5 in the main text we find a maximum contribution of roughly 10 % for $\alpha$-pinene experiments and roughly 5 % for $\beta$-caryophyllene. For most of the growth rates determined, this coagulation effect is of

the order of 1- 2 %. Further note that the numerical model proposed by McMurry (1980) is based on a collision-controlled particle formation regime suggesting rather high particle concentrations which, at least in the range above 3 nm, are not found experimentally. Thus the GR shown by Fig. 9 (b) can be seen as an estimate on the upper limit of the contribution of sub-3 nm particle to the growth rate.





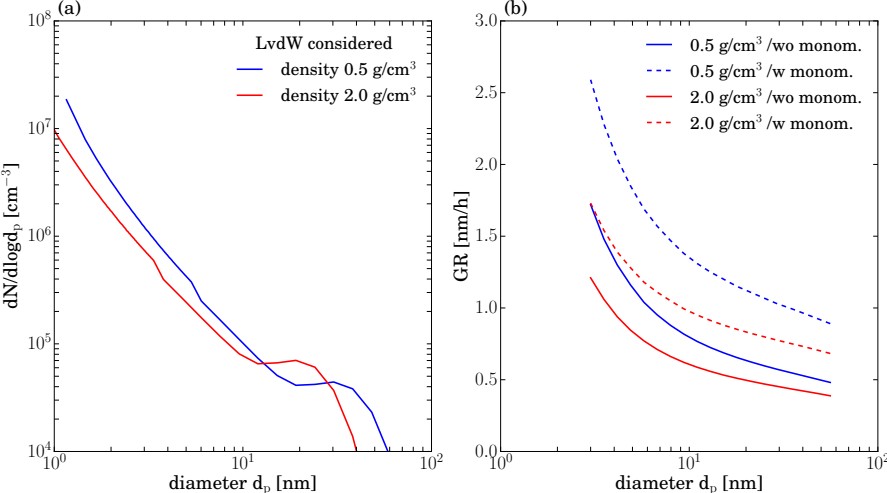

**Figure 9.** Effects of sub-$d_{min}$ particles. (a) Number size-distribution derived from dimensionless results calculated for a collision-controlled limit according to (McMurry and Li, 2017). LvdW stands for London-van-der-Waals forces, which are taken into account in calculating the resulting size-distributions. (b) GR [nm/h] resulting from collisions with sub d min particles as a function of particle diameter for particle density of 0.5 g/ccm and 2 g/ccm. Further, collisions with monomers are considered (/w monom.) and neglected (/wo monom.)

## Appendix E: Data Inversion for DMA-train and SMPS

For the experimental data obtained at the NCAR biogenic aerosol chamber a combined data inversion procedure for the DMA-train and the SMPS measuring the same aerosol source is applied. Both instruments rely on electrical mobility analysis done by differential mobility analysers (DMAs). While the SMPS is operating one DMA in scanning mode, the DMA-train operates five DMAs in parallel at fixed voltages

and hence particle sizes. Data inversion is based on the procedure of Stolzenburg and McMurry (2008):

$$\frac{\mathrm{d}N}{\mathrm{d}\ln d_p}\bigg|_{d_p^*} = \frac{N \cdot a^*}{\beta \cdot f_c(d_p^*) \cdot \eta_{\mathrm{sam}}(d_p^*) \cdot \eta_{\mathrm{cpc}}(d_p^*) \cdot \eta_{\mathrm{dma}}(d_p^*)}, \tag{E1}$$

where $N$ is the measured raw concentration downstream of the DMA, which is operated to select a centroid diameter $d_p^*$, $\beta$ is the ratio of aerosol to sheath flow in the DMA, $a^* = (-\mathrm{d}\ln Z/\mathrm{d}\ln d_p)|_{d_p^*}$, $f_c$ is the charging efficiency for singly charged particles and $\eta_{\mathrm{sam}}$ describes the total sampling losses, $\eta_{\mathrm{cpc}}$ the condensation particle counter's activation efficiency and $\eta_{\mathrm{dma}}$ the inlet- and outlet-penetration efficiencies

of the used DMA.

Note that for both instruments most of the parameters are distinct. Sampling efficiencies are inferred from sampling line lengths, sample flow rates and classified diameters and assumed to follow the diffusional losses according to Gormley and Kennedy (1948), which are different for the SMPS system and the DMA-train. CPC activation curves $\eta_{\mathrm{cpc}}$ depend on the used particle counters, a TSI Inc. Model 3760 for the SMPS , and four TSI Inc. Model 3025A CPCs and one modified TSI Model 3775, which uses diethylene glycol as working fluid

(Iida et al., 2009; Jiang et al., 2011b), for the DMA-train. DMA penetration efficiencies $\eta_{\mathrm{dma}}$ differ as well, as the DMA-train used five TSI Model 3085A nano-DMAs and the SMPS used one long column DMA, TSI Model 3081.

Moreover, Eq. E1 only considers singly charged particles. Bipolar charging probabilities below 100 nm are generally dominated by singly charged particles (Fuchs, 1963). Only a doubly charge correction was therefore applied for the SMPS data. With the SMPS data



fully corrected, it could be used to calculate the expected doubly charged contribution on the raw DMA-train signal by considering the different losses for the DMA-train system. We find that the contribution of doubly charged particles in the DMA-train measurement range is compeletly negigible in the two considered experiments.

In both of the above presented measurements, the SMPS measured down to 10 nm in size and at least one DMA-train channel was fully overlapping with the measured size-distribution of the SMPS. In the overlapping size-channel at 14 nm small deviations (<20 %) between the instruments were found. This is most probably due to uncertainties in the input of the inversion procedure, e.g. material dependencies in applied CPC counting efficiencies (Kangasluoma et al., 2014) and deviations from the assumed sampling penetration efficiencies through the usage of bended tubing (Wang et al., 2002). Therefore the SMPS spectra were normalised to fit perfectly in the overlapping region. This procedure might however cause some uncertainties in the presented analysis.

Furthermore, the DMA-train does not rely on a scanning procedure and therefore acquires concentration data at the fixed sizes within each second. The SMPS requires 120 seconds to scan from a low voltage to a high voltage and another 120 seconds to reverse. The results from each $d_{\mathrm{p}}$ bin are averaged resulting in a complete size distribution every 240 seconds. The DMA-train data were then averaged over the scanning cycle of the SMPS. This basically provided the necessary counting statistics for the DMA-train channels below 10 nm where particle penetration and charging efficiencies are usually very low. For the case of $\alpha$-pinene two runs under similar conditions (same ozone and $\alpha$-pinene concentrations) were performed and averaged in oder to improve the quality of the measured size-distributions.



## Appendix F: Errors of the analysis methods

In the following potential errors of the analysis methods TREND and INSIDE are discussed. Note that errors originating from the experiment are not part of this section which solely describes the error caused by the analysis methods itself. Both analysis methods are not exact as they are derived from quantities that are either averaged (with respect to time and/or particle diameter) or generated by means of numerical

simulation.

Further, both methods rely on simulated particle size distributions. In case coagulation is not dominant (as it is the case in the present work) the error due to numerical simulation can be neglected. Other simulation errors may originate from dilution of the aerosol and particle wall losses. Given that these processes are known (i.e. determined experimentally), the simulation result is of the order of 0.1 %.

An additional source of uncertainty is the fact that particle growth cannot be taken into account for the calculation of other dynamic

processes since it is determined from the simulated data. This affects coagulation and wall losses. In order to estimate the effect of particle growth on the calculation of wall loss, the change in median diameter determined by the TREND method is considered. For $\alpha$-pinene we find an average change ($\Delta d_{\mathrm{p}}$) of 34% with a standard deviation of 14%. $\beta$-caryophyllene shows a change of 46% with a standard deviation of 25%. Thus, the software underestimates the particle diameter and, as a consequence, overestimates deposition. To estimate the effect on the growth rate, the simulations are repeated with an altered wall loss mechanism: for the determination of wall loss the particle diameters

are increased by $\Delta d_{\mathrm{p}}$. The resulting average change in the GR is less than 2% for $\alpha$-pinene less than 4% for $\beta$-caryophyllene.

Considering inter-particle forces (Chan and Mozurkewich, 2001) enhancing coagulation results in an average error of less than 2% for both experimental data sets (note that an increase of coagulation coefficient due to inter-particle forces by a factor of 5 was assumed).

INSIDE and TREND determine growth rates for a certain time interval $\Delta t$ which limits the time-resolution. To estimate the resulting relative error we consider the growth rate at a certain diameter and at various points in time:

$$\Delta \mathrm{GR}(d_{\mathrm{p}}, t_i) = \left[ 1 - \frac{GR(d_{\mathrm{p}}, t_{i-1}) + GR(d_{\mathrm{p}}, t_{i+1})}{2 \cdot GR(d_{\mathrm{p}}, t_i)} \right] \cdot \frac{2 \cdot \Delta t}{t_{i+1} - t_{i-1}} \tag{F1}$$

The mean resulting error and corresponding standard deviation are 0.4% and 7.6%, respectively, for the $\alpha$-pinene data; and 7.2% and 26.4% for $\beta$-caryophyllene.

The TREND method calculates growth rates for regions of the number-size-distribution $[d_{\mathrm{p}}; d_{\mathrm{p}} + \Delta d_{\mathrm{p}}]$. The wider the region, the larger is the diameter range the GR is attributed to. In the present work the number-size-distribution is typically divided into 50 or 100 regions

each containing 1/50 or 1/100 of the total particle number concentration. According regions have an average width of 2% and a standard deviation of 5% for the two experimental data sets considered. In case the number of regions is getting large the TREND method approaches the INSIDE method and the error vanishes.

To conclude, for experiments with the $\alpha$-pinene and $\beta$-caryophyllene, the main error regarding the growth rate results from the choice of time period between two determinations of the growth rates. Further, the choice of the width of the regions in the TREND method is

important. Both quantities can be reduced to limit the errors. However, note that a reduction increases error due to scattering inputs from the measurement system. The only error which cannot be influenced is the error due to numerical simulation which is typically negligible ($\sim$0.1%) in case coagulation does not play a dominant role.



## Appendix G: Input data for SALSA simulations

**Table 1.** Input data used to generate particle size distributions with the SALSA module. $J_3$ depicts the nucleation rate (i.e. the number concentration of particles being added to the 3 nm sized particles per second).

| Data used for | $T$ [K] | $J_3$ [cm$^{-3}$ s$^{-1}$] | $\beta_{\text{wall}}$ [s$^{-1}$] | GR [nm h$^{-1}$] | Output |
|---|---|---|---|---|---|
| Fig. 2 | 293.15 | 10 | $0.0059 \cdot d_{\text{p}}^{-1.0341}$ | $(70-5) \cdot \frac{d_{\text{p}}^{0.01}-1.0110}{1.0471-1.011}+5$ | every 120 sec |
| Fig. 8 | 293.15 | 3000 | $0.0059 \cdot d_{\text{p}}^{-1.0341}$ | $(70-5) \cdot \frac{d_{\text{p}}^{0.01}-1.0110}{1.0471-1.011}+5$ | every 120 sec |

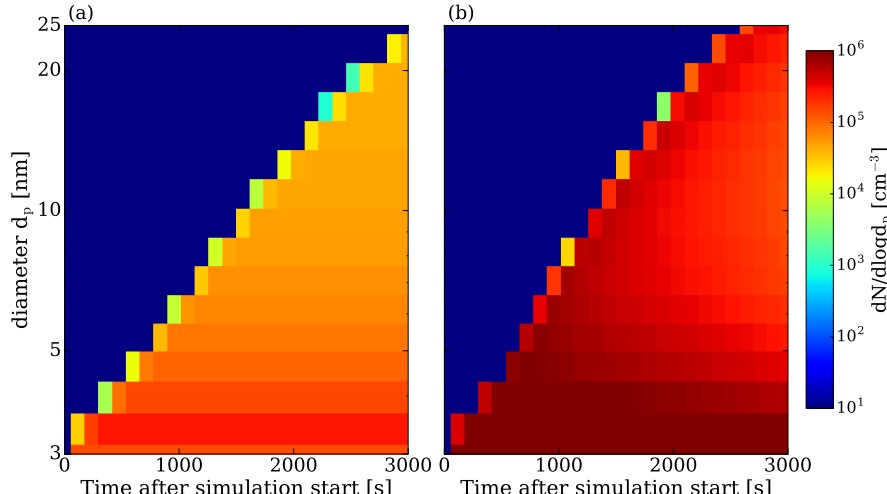

**Figure 10.** Size distributions generated by the SALSA module and used for the testing of the analysis methods in Sec. **??**. Panel (a) sows the input used for Fig. 2, while panel (b) shows the input with a higher formation rate used for Fig.8.



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
