# Peer review of "Resolving nanoparticle growth mechanisms from size- and time-dependent growth rate analysis"

_Atmospheric Chemistry and Physics, 2017_

## Referee Comment (RC1) · Anonymous Referee #1 · 20 Sep 2017

Pichelstorfer et al. proposed two methods to estimate the growth rate of sub-10 nm particles during new particle formation (NPF) events. These two independent methods are both derived from aerosol general dynamic equation. They can both provide size- and time-resolved particle growth rate. The validity of these two methods were tested using a simulated NPF event. Both method reconstructed relative good size-resolved growth rates when admitting acceptable uncertainties. These two methods together with the appearing time method were also compared in a controlled chamber study. This study is sound and solid. The reported methods serves as powerful approaches to analyze the subsequent growth after nucleation. In addition, this manuscript nicely summarized previous studies on growth rate estimation.

[Figure]

I have one major concern about the TREND method and a few minor comments to be addressed before its publication.

Major:

Although it was not clarified, essentially these is another assumption in the TREND method: particles grow monotonously by diameter such that after a short interval (tj+1 – tj), the increase caused by condensational growth can be determined by the diameter increase in the same size region. This assumption is reasonable for the simulate NPF event and the chamber study for testing and the relatively negligible effect of coagulation are evident as inferred from Appendix D. However, for NPF events in polluted atmosphere, i.e., where the coagulation scavenging effect is a major (or the dominant) factor, the feasibility of the TREND method should be carefully considered. That is, does coagulation scavenging contribute a considerable (e.g., >30%) part to the observed growth rate in urban environment (e.g., Beijing)?

Minor:

1) The determination or selection of the number of m in the TREND method should be clarified. For example, what is the exact value of m in this study? And what is the suggested range for m if provided with the particle size bins?

2) There is no need to introduce number-volume distribution in the INSIDE method. Particles are referred as volumes in the software, however, using particle diameter will help to simply the derivation. 3) It should be dN/dlogdp rather than dn/dlogdp in Eq. 9.

4) Please consider discussing why in both chamber studies the observed GR peaks around 5 nm.

5) Line 14 in page 8, the manuscript states that "good agreement is observed for all the three possible appearance time method measurements." As illustrated by Fig. 4(c), however, the GR estimated using the TREND method is more than 3 times that estimated using the appearance time method.

6) "Panel A neglects coagulation while panel B considers coagulation. (line 5 in page 15, Appendix C)" It is difficult to understand where and how coagulation was neglected.

7) Both the simulated and the observed GR are too large compared to the reported GR in the real atmosphere. The relative importance of the coagulation effect is suppressed in such conditions. Current analysis, such as those in Appendix C & D does not indicate that the coagulation scavenging effect in the real atmosphere can be adequately considered or reasonably neglected in the TREND method. The authors should clarified this.

---

## Referee Comment (RC2) · Anonymous Referee #2 · 5 Oct 2017

Review of Pichelsdorfer et al: "Resolving nanoparticle growth mechanisms from size- and time-dependent growth rate analysis".

The authors present two novel methods to derive size-dependent particle growth rate from successive particle size distribution measurements. The methods are based on analyzing changes in the particle number concentration changes as time advances. The methods are novel and interesting to the community as determining the growth rate accurately is a key question in resolving particle formation mechanisms. The methods are sound and the testing of the methods has been done rigorously and carefully. I have a few, fairly minor questions and requests for clarification regarding the manuscript; if

these can be answered, I suggest that the manuscript should be published in ACP.

Questions/requests for clarification:

* The authors compare the new methods to the appearance time (maximum concentration) method which is described as the standard protocol for ddp/dt determination in Kulmala et al., (2012). This method is not suited to produce time-dependent growth rates, although it gives a size dependence. However, the other method that has been described in the same protocol, namely the log-normal distribution function method (Kulmala et al., 2012, p.1659) can give the time-dependent growth rate of a log-normal particle population as a function of time. The authors present no comparison of their methods to this other, quite widely used growth rate determination method. I think that such a comparison could be beneficial, and should not be too laborious as simulated PSDs are already available. Otherwise, it would be good to mention why such a comparison is not considered.

* In section 2.1, it might be helpful to keep the time indexes in the derivation at all times. If I'm reading the method correctly, the experimental size distribution at $t_{j-1}$, $n^{exp}(t_{j-1})$ is taken as the starting point for producing the simulated distribution $n^{sim}(t_j)$, and this simulated distribution is then compared piecewise to the experimental distribution $n^{exp}(t_j)$. Differences are attributed to growth. From this, I would consider that the time period for which the growth is calculated is $[t_{j-1}, t_j]$. However, In equation (4), the time points used are $t_j$ and $t_{j+1}$, which is confusing, especially as the times are not given for the (time-dependent) count median diameters. Using the time indexes consistently in Eqs. (2), (3) and (4) would make this much easier to read for a reader trying to implement the method. Also, in the appendix A the methods is described as 'forward-looking', i.e. the interval $[t_j, t_{j+1}]$ is considered.

* Regarding the times, Eq. (4) should include the time stamp similarly to Eq (9).

* Is there a reason why the TREND method looks backward from $t_j$ while INSIDE looks forward?
* Section 2.2. line 17: to me it seems that Eq (5) still gives the number concentration but in volume space (change '...in integrated volume concentration...' to 'in integrated nubmer concentration')

p. 4, line 25: Dilution losses should also be mentioned as they are explicitly in Eq. (5)

p. 5: I do not fully understand why the INSIDE methods needs to use the number volume distribution instead of the number size distribution. In seems unnecessarily confusing especially as it is later converted to size anyway. Could this be explained?

p.5 line 27: '...the more pronounced the differences between the models become.' Which models? Only one model was used?

p. 6:, line 13: '...sensitive to scatter...' what is meant by scatter here? Experimental error in the size distribution or something else? Please clarify.

p10, line 7: 'This trend suggests...' This is confusing, as it reads like lower growth rates in the sub-3nm range indicate lower-volatility vapours. I think the trend meant here is the 'somewhat less significant' part of the sentence; the manner of expressing this is confusing and could be re-structured.

p. 11 line 7: '...sufficient time-resolution and appropriate counting statistics'. I did not find any criteria for this statement. The error was not analysed as a function of time resolution or counting statistics. Please give some more information on what is 'sufficient / appropriate', otherwise reword.

References

Kulmala et al., 2012, Nature Protocols 7, 1651–1667 (2012) doi:10.1038/nprot.2012.091

---

## Author Response (AR1)

**Authors Response to the Referee Comments to the manuscript "Resolving nanoparticle growth mechanisms from size- and time-dependent growth rate analysis"**

We appreciate the comments by the referees which have certainly helped improve clarity and overall quality of our manuscript. Below the replies to all points raised can be found.

**Response to Anonymous Referee #1:**

General:

*"Although it was not clarified, essentially there is another assumption in the TREND method: particles grow monotonously by diameter such that after a short interval ($t_{j+1} - t_j$), the increase caused by condensational growth can be determined by the diameter increase in the same size region. This assumption is reasonable for the simulate NPF event and the chamber study for testing and the relatively negligible effect of coagulation are evident as inferred from Appendix D. However, for NPF events in polluted atmosphere, i.e., where the coagulation scavenging effect is a major (or the dominant) factor, the feasibility of the TREND method should be carefully considered. That is, does coagulation scavenging contribute a considerable (e.g., >30%) part to the observed growth rate in urban environment (e.g., Beijing)?"*

We acknowledge this detailed comment about the effect of coagulation and background aerosol effects. Generally we have to underline, that both models do not assume monotonous growth of particles within a time interval as they both consider the contribution of coagulation, condensational growth, wall losses and dilution losses. However, the condensational growth (described by means of growth rate in the manuscript) is assumed to be constant during this time interval. This might lead to limitations of the models, which are listed in the manuscript in section 2.1 (TREND) and in section 2.2 (INSIDE). A potential source of error remains through the contribution of sub-detection limit particles which seemed to be negligible for the considered experiment (see appendix D), but could play a more decisive role in highly polluted areas. Moreover, it is true that the currently used model does not consider background aerosol. Theoretically, this could be included into the model as long as the background aerosol and the particles originating form NPF can be separated. The scavenging caused by coagulation is not believed to cause severe problems as coagulation losses are already included into the model. This is done by using the widely accepted approach of (Fuchs & Sutuguin, (1971). *Highly dispersed aerosols*. New York: Pergamon Press.) to describe the coagulation kernel in the transition regime. To summarize, the scavenging effect caused by coagulation of newly formed particles onto a background aerosol, no matter what concentration, can theoretically be considered by both analysis methods (TREND and INSIDE), which is work-in-progress but will be published separately. However, we certainly see the need to clarify the importance of this effect and included an additional paragraph on **page 10 line 26-29.**

Minor:

*"The determination or selection of the number of m in the TREND method should be clarified. For example, what is the exact value of m in this study? And what is the suggested range for m if provided with the particle size bins?"*

We already gave the numbers for m in Appendix F, but agree that we should specify this more precisely and modified this paragraph on **page 19 line 23-28**.

*"There is no need to introduce number-volume distribution in the INSIDE method. Particles are referred as volumes in the software, however, using particle diameter will help to simplify the derivation."*

We changed the volume dependent description in section 2.2 to a diameter dependent description. Changes can be found on **page 4 line 20 – page 5 line 13**.

*"It should be dN/dlogdp rather than dn/dlogdp in Eq. 9."*

We replaced dn/dlogdp by dN/dlogdp in Eq. 9 on **page 5 line 11** and **page 5 line 18**.

*"Please consider discussing why in both chamber studies the observed GR peaks around 5 nm."*

In Section 5 we explained in detail the appearance of a multi-component Kelvin effect causing an increase in particle growth rates towards larger diameters. Tröstl et al. (2016) explained this effect in detail and found as well a small maximum around 10 nm for simulated growth rates from an assumed volatility distribution. Another possibility for a maximum in growth rates could be diameter dependent particle phase reactions and resulting reactive uptake. To answer this question for our study we would have needed more detailed measurements of

particle and gas phase composition. We therefore do not want to speculate too much about the observed peak but rather want to show that sub-10 nm growth shows more variety than generally believed and we therefore want to draw the focus on more detailed and comprehensive measurements in this regime, which we have stated as the final conclusion of our manuscript.

*"Line 14 in page 8, the manuscript states that "good agreement is observed for all the three possible appearance time method measurements." As illustrated by Fig. 4(c), however, the GR estimated using the TREND method is more than 3 times that estimated using the appearance time method."*
Considering the rather large uncertainties of the appearance time method in both the time and the growth rate dimension, we think generally good agreement is achieved. We changed the statement to "The appearance time method shows a similar trend as the TREND-method for the three possible appearance time-measurements." on **page 8 line 17-18**.

*"Panel A neglects coagulation while panel B considers coagulation. (line 5 in page 15, Appendix C)" It is difficult to understand where and how coagulation was neglected."*
We changed the respective sentence on **page 15 line 5**: "Results in Panel A were calculated neglecting coagulation while for the results shown in Panel B, the effect of coagulation was included."

*"Both the simulated and the observed GR are too large compared to the reported GR in the real atmosphere. The relative importance of the coagulation effect is suppressed in such conditions. Current analysis, such as those in Appendix C & D does not indicate that the coagulation scavenging effect in the real atmosphere can be adequately considered or reasonably neglected in the TREND method. The authors should clarified this."*
We appreciate this thoughtful comment that coagulation effects are more important in the case of low GRs and might therefore be more important when analyzing ambient data. We still think, that Appendix C shows quite well, that we already can handle coagulation within our models to some extent. Moreover, as included into our statement to the major comment, we modified our conclusions on **page 10 line 26-29** including a statement that a more detailed description of coagulation is work-in-progress. Moreover, we added a statement in Appendix C, **page 15 line 9-12**, pointing out the need of more sophisticated coagulation tests when the framework is applied to ambient data with low GRs and high coagulation sinks.

**Response to Anonymous Referee #2:**

*"The authors compare the new methods to the appearance time (maximum concentration) method which is described as the standard protocol for ddp/dt determination in Kulmala et al., (2012). This method is not suited to produce time-dependent growth rates, although it gives a size dependence. However, the other method that has been described in the same protocol, namely the log-normal distribution function method (Kulmala et al., 2012, p.1659) can give the time-dependent growth rate of a log-normal particle population as a function of time. The authors present no comparison of their methods to this other, quite widely used growth rate determination method. I think that such a comparison could be beneficial, and should not be too laborious as simulated PSDs are already available. Otherwise, it would be good to mention why such a comparison is not considered."*
We thank the referee for pointing out that we did not mention the lognormal distribution function, when we were listing the GR calculation methods. However, as can be seen from our size-distribution data shown in Fig. 3, Fig. 4 and Fig. 10, our nucleation mode does not follow a classical lognormal distribution and is therefore difficult to fit, which we mention now on **page 9 line 8-9**. This is observed for other chamber experiments as well, i.e. compare with the size-distribution data from Töstl et al., 2016, Nature. We added a statement on the log-normal distribution function method in the introduction (**page 2, line 6-7).** Further we analyzed our data with this method where measured size distribution could reasonable be approximated by a lognormal function (see **Figure 5** and **page 9 line 7**).

*"In section 2.1, it might be helpful to keep the time indexes in the derivation at all times. If I'm reading the method correctly, the experimental size distribution at $t_{j-1}$, $nexp(t_{j-1})$ is taken as the starting point for producing the simulated distribution $nsim(t_j)$, and this simulated distribution is then compared piecewise to the experimental distribution $nexp(t_j)$. Differences are attributed to growth. From this, I would consider that the time period for which the growth is calculated is $[t_{j-1}; t_j]$. However, In equation (4), the time points used are $t_j$ and $t_{j+1}$, which is confusing, especially as the times are not given for the (time-dependent) count median diameters. Using the time indexes consistently in Eqs. (2), (3) and (4) would make this much easier to read for a reader trying to implement*

*the method. Also, in the appendix A the methods is described as 'forward-looking', i.e. the interval [$t_j$ ; $t_{j+1}$] is considered."*
We changed the equations, figures and text to have a uniform 'forward looking' representation on **page 2 line 28** until **page 4 line 5**.

*"Regarding the times, Eq. (4) should include the time stamp similarly to Eq (9)."*
We changed equation 4 to include the time stamp on **page 4 line 8**.

*"Is there a reason why the TREND method looks backward from tj while INSIDE looks forward?"*
No, there is no reason. We changed the text and equations to have a uniform forward looking representation as mentioned earlier.

*"Section 2.2. line 17: to me it seems that Eq (5) still gives the number concentration but in volume space (change '...in integrated volume concentration...' to 'in integrated number concentration')"*
We changed it to "in integrated number concentration" on **page 4 line 20** as we changed the whole description towards number-size-distributions.

*"p. 4, line 25: Dilution losses should also be mentioned as they are explicitly in Eq. (5)"*
Dilution losses are already mentioned on **page 4 line 27 -28**: "…and the fourth term losses by dilution."

*"p. 5: I do not fully understand why the INSIDE methods needs to use the number volume distribution instead of the number size distribution. In seems unnecessarily confusing especially as it is later converted to size anyway. Could this be explained?"*
We changed the volume dependent description in section 2.2 to a diameter dependent description, see comments to Anonymous Referee #1.

*"p.5 line 27: '...the more pronounced the differences between the models become.' Which models? Only one model was used?"*
We adjusted the sentence on **page 6 line 3-5** to clarify that we mean differences between SALSA input growth rates and our growth rate results.

*"p. 6, line 13: '...sensitive to scatter...' what is meant by scatter here? Experimental error in the size distribution or something else? Please clarify."*
We specified on **page 6 line 17 – page 7 line 2** that experimental input data might suffer from statistical errors hence reducing the performance of the methods compared to the simulated input.

*"p10, line 7: 'This trend suggests...' This is confusing, as it reads like lower growth rates in the sub-3nm range indicate lower-volatility vapours. I think the trend meant here is the 'somewhat less significant' part of the sentence; the manner of expressing this is confusing and could be re-structured."*
We modified our statement on **page 10 line 11-13** to point out more clearly that it is the smaller reduction of small diameter growth rates which points towards the fact that we produce less volatile vapours.

*"p. 11 line 7: '...sufficient time-resolution and appropriate counting statistics'. I did not find any criteria for this statement. The error was not analysed as a function of time resolution or counting statistics. Please give some more information on what is 'sufficient / appropriate', otherwise reword."*
We slightly changed our statement on **page 11 line 13** and gave an approximated but more precise value for the time-resolution we assume to be necessary. We already addressed the question why limited time-resolution causes an error for the GDE evaluation in Appendix F, where even a quantification of that error is given. Moreover, we mention several times in the manuscript, that counting statistics directly impact the particle size distribution measurement and that the determination of the growth rates is the more reliable the better the measurement of the evolution of the number-size-distribution is.

[revised manuscript text omitted]
}}\left(t_{j+1}, d_p\right)$ and $n^{\text{exp}}\left(t_{j+1}, d_p\right)$. For each of the regions ($r_{i=[1,m]}$) of the experimental and the simulated number-size-distribution, the count median diameter $d_{\text{CMD},i}$ is determined and used to calculate the growth rate $GR$ of a particle with diameter $d_{p,i}^* = \frac{d_{\text{CMD},i}^{\text{exp}} + d_{\text{CMD},i}^{\text{sim}}}{2}$:

$$GR\left(d_{p,i}^*\right) = \frac{d_{\text{CMD},i}^{\text{exp}} - d_{\text{CMD},i}^{\text{sim}}}{t_{j+1} - t_j}$$

$$GR\left(d_{p,i}^*, (t_{j+1} + t_j)/2\right) = \frac{d_{\text{CMD},i}^{\text{exp}} - d_{\text{CMD},i}^{\text{sim}}}{t_{j+1} - t_j} \tag{4}$$

Note that the procedure described above has limitations and benefits of which the most important are listed below:

1. Growth is decoupled from the other dynamic processes. Thus large relative changes in the region limits $d_{p,r_i}$ may cause errors.

2. Rapid changes in the growth rate require adequate time resolution of the experimental data as the result of the analysis method is a mean growth rate for the respective time interval.

3. Influence of the coagulation process by particles smaller than $d_{\text{p,min}}$ can only be estimated.

4. The present method utilizes integral values to determine the growth rate. Thus local minima and maxima of the measured number-size-distribution (e.g. due to low particle concentration) may cancel out. However, this depends on the choice of the width of regions, which can be set for each analysis run.

**2.2 *IN*terpreting the change rate of the *S*ize-*I*ntegrated general *D*ynamic *E*quation: the INSIDE method**

The second method is based on explicit manipulation of the adapted, size-integrated GDE (see Eq. 1) which gives the change in integrated

$$\frac{dN_\infty}{dt}\Bigg|_{v_{\text{eval}}}^{\infty} = \frac{dv}{dt}\Bigg|_{v_{\text{eval}}} \cdot n(v,t)\Big|_{v_{\text{eval}}} + \frac{dN_{\infty,coag}}{dt} - \int_{v_{\text{eval}}}^{\infty} \beta_{\text{wall}}(v)\,n(v,t)\,dv - \beta_{\text{dil}}N_\infty$$

5  number concentration featuring a diameter larger than $d_{\text{eval}}$:

$$\frac{dN_\infty}{dt}\Bigg|_{d_{\text{eval}}}^{\infty} = \frac{dd_{\text{p}}}{dt}\Bigg|_{d_{\text{eval}}} \cdot n(d_{\text{p}},t)\Big|_{d_{\text{eval}}} + \frac{dN_{\infty,coag}}{dt} - \int_{d_{\text{eval}}}^{\infty} \beta_{\text{wall}}(d_{\text{p}})\,n(d_{\text{p}},t)\,dd_{\text{p}} - \beta_{\text{dil}}N_\infty \tag{5}$$

where $t$ is time, $v$ is the particle volume,  $n(d_{\text{p}},t)$ is the number-size-distribution and $d_{\text{eval}}$ the smallest particle  diameter considered (not necessarily equal to the minimum measured  diameter $d_{\text{p,min}}$). $N_\infty$ depicts the total integrated number concentration, from  $d_{\text{eval}}$ to $\infty$. Note that compared to Eq. 1, the loss-

10  terms have been adopted for a chamber-experiment and the particle current  $I(d_{\text{p}})$ now represents the particle growth at the evaluation  size $d_{\text{eval}}$.

The first term on the right hand side considers particles that grow into the range [ $d_{\text{eval}}$, $\infty$]; the second term considers  number-size-distribution changes due to coagulation; the third term describes losses at system walls; and the fourth term losses by dilution. Coagulation and wall losses are approximated by a comparison between the

15  simulated number-distribution $n^{\text{sim}}(t_{j+1})$ and the experimental number-distribution $n^{\text{exp}}(t_j)$ at two discrete and subsequent points in time and for the considered interval [ $d_{\text{eval}}$,$\infty$]:

$$\frac{dN_\infty^{\text{sim}}}{dt}\Bigg|_{v_{\text{eval}}}^{\infty} = \frac{dN_{\infty,coag}}{dt}\Bigg|_{v_{\text{eval}},\infty} - \int_{v_{\text{eval}}}^{\infty} \beta_{wall}(v)\,n(v,t)\,dv - \beta_{\text{dil}}N_\infty \approx \frac{N_{[v_{\text{eval}},\infty]}^{\text{sim}}(t_{j+1}) - N_{[v_{\text{eval}},\infty]}^{\text{exp}}(t_j)}{t_{j+1}-t_j}$$

$$\frac{dN_\infty^{\text{sim}}}{dt}\Bigg|_{d_{\text{eval}}}^{\infty} = \frac{dN_{\infty,coag}}{dt}\Bigg|_{d_{\text{eval}},\infty} - \int_{d_{\text{eval}}}^{\infty} \beta_{wall}(d_{\text{p}})\,n(d_{\text{p}},t)\,dd_{\text{p}} - \beta_{\text{dil}}N_\infty \approx \frac{N_{[d_{\text{eval}},\infty]}^{\text{sim}}(t_{j+1}) - N_{[d_{\text{eval}},\infty]}^{\text{exp}}(t_j)}{t_{j+1}-t_j} \tag{6}$$

20  Therefore equation (5) can be rearranged:

$$\frac{dv}{dt}\Bigg|_{v_{\text{eval}}} = \frac{\dfrac{dN_\infty}{dt}\Big|_{v_{\text{eval}}}^{\infty} - \dfrac{dN_\infty^{\text{sim}}}{dt}\Big|_{v_{\text{eval}}}^{\infty}}{n(v,t)\big|_{v_{\text{eval}}}}$$

$$\frac{dd_{\text{p}}}{dt}\Bigg|_{d_{\text{eval}}} = \frac{\dfrac{dN_\infty}{dt}\Big|_{d_{\text{eval}}}^{\infty} - \dfrac{dN_\infty^{\text{sim}}}{dt}\Big|_{d_{\text{eval}}}^{\infty}}{n(d_{\text{p}},t)\big|_{d_{\
[revised manuscript text omitted]